# Missense variant interaction scanning reveals a critical role of the FERM domain for tumor suppressor protein NF2 conformation and function

Christina S Moesslacher[1], Elisabeth Auernig[1], Jonathan Woodsmith[1], Andreas Feichtner[2], Evelyne Jany-Luig[1], Stefanie Jehle[3], Josephine M Worseck[3], Christian L Heine[1], Eduard Stefan[2,4,5], Ulrich Stelzl[1,3,6,7]

**NF2 (moesin–ezrin–radixin-like [MERLIN] tumor suppressor) is frequently inactivated in cancer, where its NF2 tumor suppressor functionality is tightly coupled to protein conformation. How NF2 conformation is regulated and how NF2 conformation influences tumor suppressor activity is a largely open question. Here, we systematically characterized three NF2 conformation-dependent protein interactions utilizing deep mutational scanning interaction perturbation analyses. We identified two regions in NF2 with clustered mutations which affected conformation-dependent protein interactions. NF2 variants in the F2–F3 subdomain and the α3H helix region substantially modulated NF2 conformation and homomerization. Mutations in the F2–F3 subdomain altered proliferation in three cell lines and matched patterns of disease mutations in NF2 related-schwannomatosis. This study highlights the power of systematic mutational interaction perturbation analysis to identify missense variants impacting NF2 conformation and provides insight into NF2 tumor suppressor function.**

## Introduction

NF2 (moesin–ezrin–radixin-like [MERLIN] tumor suppressor) is a member of the band 4.1 superfamily of proteins and is closely related to ezrin–radixin–moesin (ERM) proteins, which functions as links between the cell membrane and actin filaments (Bretscher et al, 2002; Laulajainen et al, 2008). ERM protein family members have been linked to cancer (Clucas & Valderrama, 2014); however, NF2 tumor suppressor activity was initially characterized in flies and mice (Rouleau et al, 1993; Trofatter, 1993), and then in mammalian cell models. NF2 links signals from the cell membrane to growth-related gene expression and acts in cell–cell contact inhibition (Morrison et al, 2001; Okada et al, 2005; Curto et al, 2007), a function defined as one of the hallmarks of cancer. In contrast to the other ERM family members, NF2 lacks an F-actin-binding domain. It also binds to phosphatidylinositol lipids (Okada et al, 2009). NF2 is found at the cell membrane in contact with CD44, where it organizes cell junctions (Lallemand et al, 2003) and growth factor receptors (Curto et al, 2007; Lallemand et al, 2009). As an important regulator of cell growth, NF2 impacts proliferation-associated pathways such as MAPK, AKT, and Rac signaling (Morrison et al, 2007; Cui et al, 2019). It also directly modulates transcription cofactor regulation, via AMOT/LATS or DCAF1, in the YAP/TAZ-hippo pathway (Hamaratoglu et al, 2006; Li et al, 2010; Zhang et al, 2010; Cooper & Giancotti, 2014). More recently, NF2 was found to be an upstream factor of nucleic acid sensing, suppressing cGAS-STING–initiated antitumor immunity in cancer cell models (Meng et al, 2021).

Genetics also defines a prominent role of *NF2* loss of function in cancer. Genetic mutations or deletions of *NF2* cause NF2-related schwannomatosis (Plotkin et al, 2022), an autosomal dominant disease predisposing to the formation of benign tumors. Biallelic *NF2* mutations cause tumor formation in the nervous system represented by vestibular schwannomas, meningiomas, and ependymomas, frequently accompanied by hearing loss, dizziness, and neuropathies (Asthagiri et al, 2009; Evans, 2009). *NF2* mutations are also commonly found in aggressive malignant mesothelioma, and frequently observed in other cancer types such as melanoma, breast, prostate, liver, and kidney cancers (Petrilli & Fernández-Valle, 2016; Martincorena et al, 2017). *NF2* is a general tumor suppressor and cancer driver gene affected by mutations (tumor suppressor gene score 89% in pan cancer analysis [Vogelstein et al, 2013]).

NF2 protein is expressed in various splice isoforms with the canonical isoform 1 being the longest (Figs 1A and S1A). Isoform 2 has the same N-terminal part as isoform 1 and differs from isoform 1 only at the C-terminus through including exon 16 instead of exon 17. Ending with amino acid sequences from exon 16, the C-terminus

[1]Institute of Pharmaceutical Sciences, Pharmaceutical Chemistry, University of Graz, Graz, Austria    [2]Institute of Biochemistry and Center for Molecular Biosciences, University of Innsbruck, Innsbruck, Austria    [3]Max-Planck Institute for Molecular Genetics (MPIMG), Otto-Warburg-Laboratory, Berlin, Germany    [4]Tyrolean Cancer Research Institute (TKFI), Innsbruck, Austria    [5]Institute of Molecular Biology, Innsbruck, Austria    [6]BioTechMed-Graz, Graz, Austria    [7]Field of Excellence BioHealth - University of Graz, Graz, Austria

Correspondence: ulrich.stelzl@uni-graz.at

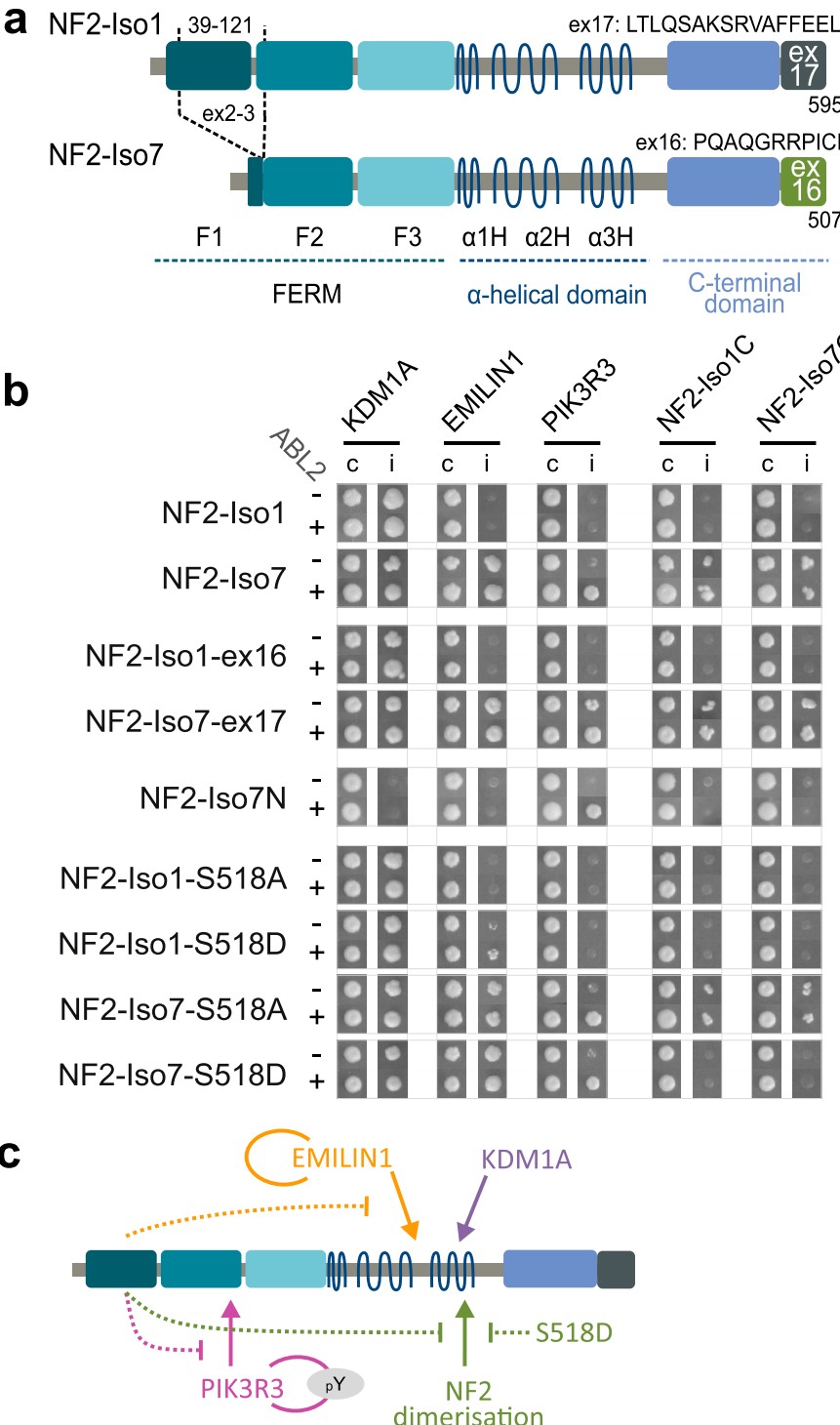

**Figure 1. Conformation dependent PPIs of NF2.**
**(A)** NF2 splice isoforms. The canonical isoform 1 is 595 amino acids long, the C-terminus is derived from sequences of exon 17 (595 AA [ex17], P35240-1). NF2 isoform 7 is a shorter splice isoform (507 AA [Δex2/3, ex16], P35240-4). The C-terminus is alternatively spliced and includes exon 16-derived sequences which are 5 amino acids shorter (iso-1 580–590: LTLQSAKSRVA → PQAQGRRPICI, iso-1 590–595 missing), isoform 7 lacks most of the F1 FERM subdomain and the beginning of the F2 subdomain encoded in exons 2 and 3, amino acids 39–121. Tumor suppressor activity has been demonstrated for isoform 1 but not for isoform 7. **(B)** Y2H protein interaction results. Indicated NF2 bait constructs (row, see Fig S1A) were tested with five different prey constructs (columns) in the presence and absence of active tyrosine kinase ABL2 (+) and ABL2KD (K35M) (−) respectively. Note: NF2-Iso1N (1–332) was autoactive and was excluded from the analysis. Prey: KDM1A (O60341), EMILIN1 (Q9Y6C2), PIK3R3 (Q92569), NF2-Iso1C (P35240-1: 308–595), NF2-Iso7C (P35240-4: 225–507). Growth of diploid yeast on non-selective agar (c) and on selective agar indicating protein interactions (i) is shown. **(C)** Graphical summary of the interactions observed. Interactions with NF2 are shown with arrows; conformational interaction inhibition is indicated with dashed lines.

of isoform 2 and 7 is five amino acids shorter than in isoform 1. In comparison to isoforms 1 and 2, isoform 7 skips exons 2 and 3 and therefore lacks N-terminal amino acids 39–121 corresponding to most of the F1 and the initial amino acids of the F2 FERM subdomain.

The isoform 1 has tumor suppressor activity, whereas this activity was not demonstrated for all other NF2 isoforms including isoforms 2 and 7. Expression of isoform 2 was insufficient to impair growth in RT4-D6P2T cells (Sherman et al, 1997; Scoles et al, 2002); however, in HEI-193 cells, the stable expression of isoform 2 resulted in a

reduction in relative BrdU-levels that was comparable with isoform 1 expression (Sher et al, 2012). It was shown that deletion of exon 2 or exons 2–3 resulted in a dislocation of NF2 from the plasma membrane (Deguen et al, 1998; Koga et al, 1998). A transgenic mouse model revealed promotion of Schwann cell proliferation by a NF2 mutant lacking amino acids 39–121, which was not observed for a C-terminal-truncated NF2 isoform 1 protein version (Giovannini et al, 1999). In *D. melanogaster*, deletion of a seven amino acid-long stretch in F2 (AA 177–183 in HsNF2) was dominant negative for NF2 growth control (LaJeunesse et al, 1998) and a mutant construct with these seven amino acids replaced to alanine resulted in transformation and uncontrolled proliferation of cultured murine fibroblasts (Johnson et al, 2002). A splice isoform lacking exons 2–4 detected in hepatocellular carcinoma cell lines was unable to suppress cell proliferation (Luo et al, 2015). Based on the literature, we conclude that isoform 1 has tumor suppressor activity, whereas isoform 7 likely represents a version lacking tumor suppressor activity (Figs 1A and S1).

The importance of protein conformation for the tumor suppressor function of NF2 was recognized in early experiments (Sherman et al, 1997). Analogous to other ERM proteins, conformational rearrangements within NF2 are associated with functional changes, which are presumably triggered by posttranslational modifications (PTMs), lipid- and protein-binding. NF2 can switch between an open and a closed conformation by self-association. The C-terminal residues of NF2 and a fully folded FERM domain are thought prerequisite for the formation of the head-to-tail interaction (Grönholm et al, 1999; Gutmann et al, 1999), membrane localization (Brault et al, 2001), and tumor suppressor function (Sherman et al, 1997). However, alternative conformations have been proposed as well and the exact mechanism how NF2 conformation is regulated and how NF2 conformation influences tumor suppressor activity remains elusive (Petrilli & Fernández-Valle, 2016).

For example, S518 is a critical phospho-site in NF2 and was suggested to influence signaling activity, localization, protein interaction, and NF2 conformation. PAK2 (Gene ID: 5062) and PKA (Gene ID: 5566) were implicated in phosphorylation of S518 (Kissil et al, 2002; Xiao et al, 2002; Alfthan et al, 2004). Most of the data suggest reduced tumor suppressive function caused by S518 phosphorylation. For example, in one study, the phosphorylation mimicking S518D mutation in the full-length and the C terminal constructs prevented association with the N terminal construct, and the WT and S518A mutations allowed interaction with the N terminal part (Shaw et al, 2001). Conversely, in another study C-terminal NF2 constructs containing S518 phosphorylation mimicking mutations S518D and S518E increased the binding to N-terminal protein fragments, whereas the S518A amino acid exchange abolished an intramolecular interaction in co-IP experiments in HEI 193 cells (Sher et al, 2012). Alternative models considering various degrees of conformational open- and close-ness are put forward to reconcile a series of apparently contradicting results (Hennigan et al, 2010; Sher et al, 2012). The interpretation of the data is complicated by experimental difficulties to monitor protein conformation and to clearly distinguish intra and inter–NF2 interactions. NF2 homodimerizes at various degrees (Grönholm et al, 1999; Meng et al, 2000; Phang et al, 2016), and recently, it was

shown that gain of function mutations can even promote the formation of NF2 cellular condensates in conjunction with IRF3 (Meng et al, 2021). Elucidating NF2 conformation and its effect on interaction partners and protein function remains a pivotal research task to better understand NF2 tumor suppressor activities.

Mutational scanning (DMS) has emerged as a powerful approach to systematically map amino acid residue activity landscapes of proteins under defined readouts, yielding insights into protein function, structure, and evolution (Moesslacher et al, 2021). At the same time, these data assist computational variant effect prediction and clinical variant interpretation (Esposito et al, 2019). Physical protein-protein interactions (PPIs) are critical to perhaps all biological processes and as such PPIs are basic cellular functional units that can be assayed universally for, in principle, all proteins (Woodsmith et al, 2017; Yadav et al, 2020). We developed a deep mutational scanning protein interaction perturbation screening technique based on reverse yeast two-hybrid (Y2H) analysis (Woodsmith et al, 2017). Here, we used this method to scan a comprehensive set of single amino acid NF2 variants using four conformation-dependent PPIs as readout. This allowed to investigate the mutational impact on NF2 protein conformational regulation and thus revealed amino acid residues critical for NF2 function.

## Results

### Conformation dependent NF2–protein interactions

We tested tumor suppressive NF2 isoform 1 and the shorter non-tumor suppressive isoform 7 (Fig 1A) for protein interactions and found three isoform-specific NF2 PPIs in a phospho-Y2H screen, that involves an active protein kinase to additionally detect protein interactions that are modulated by phosphorylation (Grossmann et al, 2015; Jehle et al, 2022). The Y2H experiments revealed that lysine-specific histone demethylase 1A, KDM1A, elastin microfibril interfacer 1 protein, EMILIN1, and phosphoinositide-3-kinase regulatory subunit 3, PIK3R3, interacted with NF2 (Fig 1B). The NF2–KDM1A (Weimann et al, 2013; Haenig et al, 2020; Go et al, 2021) and NF2–EMILIN (Haenig et al, 2020) interactions were listed in other systematic large-scale PPI studies, however were not characterized any further. Here, we report isoform-specific interactions with NF2. KDM1A equally interacted with NF2 isoform 1 and isoform 7, EMILIN1 interacted strongly with NF2 isoform 7, but not with isoform 1. Both PPIs were not affected by the presence of an active or a kinase-dead version of non-receptor tyrosine kinase ABL proto-oncogene 2 (ABL2). In the case of PIK3R3, co-expression of active ABL2 was required for the interaction with NF2 isoform 7. PIK3R3 formed homodimers, and notably this PIK3R3 homodimerization was a pY-dependent interaction (Fig S1B). Therefore, the requirement of an active tyrosine kinase for the NF2–PIK3R3 interaction may be explained through facilitating PIK3R3 dimerization rather than a phosphorylation-dependent interaction of PIK3R3 with NF2. Similar to the NF2–EMILIN interaction, the NF2–PIK3R3 interaction was not observed with isoform 1 (Fig 1B).

We narrowed the NF2 binding site for the three interaction partners by using constructs that represented the N-terminal halves of isoform 1 (AA 1-K332, Iso1N) and 7 (AA 1-K249, Iso7N) and the C-terminal halves of isoforms 1 (AA M308-595, Iso1C) and 7 (AA M225-507, Iso7C). EMILIN1 and KDM1A bound to the C-terminal half irrespective of the actual C-terminal ex17 or ex16 sequence (Fig S1). In contrast, PIK3R3 interacted with the N-terminal part of NF2 isoform 7 (Figs 1B and S1). In summary, we found three isoform-specific protein interaction partners for NF2 through Y2H analyses—KDM1A, EMILIN1, and PIK3R3. When divided into two halves, the C-terminal part of NF2 protein interacted with KDM1A and EMILIN1, whereas PIK3R3 bound to a construct covering the isoform 7 N-terminal half of NF2. Because the longer isoform 1 contained all amino acids present in isoform 7, but does not interact with two of the three partners, we concluded that the protein interactions are sensitive to the NF2 conformation.

The very C-terminal residues of isoform 1 are thought to be necessary for the formation of a closed conformation (Sherman et al, 1997; Grönholm et al, 1999; Gutmann et al, 1999). Hence, we mutually exchanged the C-termini in isoform 1 and 7 and tested isoform 1 with an exon 16 derived - C-terminus (Iso1-ex16, identical to isoform 2) and isoform 7 with an exon 17 derived C-terminus (Iso7-ex17) for Y2H protein interactions (Fig S1A). A weak increase in Y2H growth was observed with Iso7-ex17 - PIK3R3 in the absence of active ABL2. Except for this protein pair, NF2 protein interactions with mutually exchanged C-termini were the same as their WT counterparts (Fig 1B). This suggests that the isoform-dependent interaction specificity is not dictated by the very C-terminal part of NF2. Moreover, because EMILIN binds to the C-terminal half of NF2 (Fig S1B), the PPI pattern suggest that isoform specificity of the interaction is because of features in the FERM domain.

When we tested NF2 for homomeric interaction in the Y2H assay we found that full-length isoform 7 but not isoform 1 interacted with the C-terminal half of NF2 independently of whether the very C-terminal amino acids of the NF2 fragments or the full-length NF2 partner resembled isoform 1 (ex17) or isoform 7 (ex16) (Fig 1B). The NF2 N-terminal constructs (Iso1N and Iso7N), when used as prey, did not show any interaction with full-length NF2 (Fig S1B). The NF2-Iso7N bait construct, although it interacted with PIK3R3, did not interact with either the Iso1C or Iso7C. In conclusion, this suggested, that the homodimeric interaction observed with the full-length NF2 isoform 7 is mediated by the C-terminal part. Secondly, the isoform specificity had to be explained by the N-terminal part of NF2, because isoform-specific differences in the C-terminal amino acid sequence did not influence the NF2–NF2 interaction. Therefore, the NF2-Iso1 N-terminal FERM domain apparently causes the inhibition of a C-terminally mediated NF2–NF2 interaction, consistent with the hypothesis that isoform 1 adopts a protein conformation distinct to isoform 7.

As mentioned above, different sets of experiments (Shaw et al, 2001; Sher et al, 2012) linked the phosphorylation of S518 to altered NF2 homomeric interaction and conformation using S518D phospho-mimicry and S518A phospho-null NF2 mutations. We tested the S518D and S518A (isoform 1 numbering used for variants throughout) mutant NF2 versions in Y2H assays for interactions with KDM1A, EMILIN1, PIK3R3, and NF2 itself (Fig 1B). The KDM1A interaction with NF2 was not substantially affected by changing S518 to

either D or A. We observed a weak signal for the EMILIN–NF2-Iso1-S518D interaction and reduced growth for the PIK3R3–NF2-Iso7-S518D pair in comparison with WT NF2. Interestingly, although the NF2 isoform 7 S518A mutant NF2 version, just as WT NF2-Iso7, did show homomeric NF2 interaction, S518D perturbed the NF2 interaction between mutated full-length isoform 7 and the NF2 C-terminal part (Fig 1B). Apparently, phenocopying the isoform 1 FERM domain in NF2 in our assay, we concluded that the S518D phospho-mimicry mutant version negatively affects the PIK3R3 interaction and the NF2 isoform 7 homo-interaction.

In summary (Fig 1C), our Y2H studies revealed that KDM1A interacted with NF2 isoforms 1 and 7, and EMILIN1 and PIK3R3 interacted with NF2 isoform 7 only. The PIK3R3 interaction was likely promoted through pY-dependent PIK3R3 homodimerization. We observed a homodimeric interaction between full-length isoform 7 and the C-terminal part of NF2 which was inhibited through either an isoform 1 FERM domain or a S518D phospho-mimicry mutation in full-length NF2. It is important to emphasize that isoform 1 was active in the Y2H assay and contained all features of the shorter isoform 7. Because only the shorter isoform interacted with EMILIN and PIK3R3 and formed homomeric interactions, the isoform differences must be explained by indirect effects such as intra or intermolecular interactions of NF2. We hypothesized that the conformation of NF2 is critical for the isoform-specific protein interaction patterns. Therefore, the binding status of KDM1A, EMILIN, and PIK3R3 reflects different NF2 conformations. Single amino acid point mutations that are critical for NF2 conformation may therefore be reflected in alterations of protein interaction patterns.

## Deep mutational scanning interaction perturbation analysis of NF2

We developed reverse Y2H strains that can be used to select noninteracting protein variants from complex genetic libraries (Woodsmith et al, 2017). The NF2–Y2H interactions gave robust growth repression on media lacking adenine, prerequisite for stringent reverse selection (Fig S2A). We performed deep mutational protein interaction perturbation analysis with NF2 isoform 7 and its four WT interacting proteins: KDM1A, EMILIN1, PIK3R3, and NF2 C-terminal domain. A mutagenic library of NF2 isoform 7 containing all possible single amino acid exchanges to alanine (A:: GCT), lysine (K::AAA), glutamic acid (E::GAA), and leucine (L::TTG) was generated using a multi-step PCR-based deep mutagenesis approach with on-chip-synthesized oligonucleotides (Kitzman et al, 2015). The NF2 mutagenic pool was subcloned in the bait Y2H vector and used for transformation of the reverse-Y2H strains, mated in duplicates with the four prey strains in the presence of ABL2 and ABL2-KD (Fig S2B). Interaction perturbing NF2 variants for a given partner were enriched through growth on selective agar lacking adenine and disruptive mutants were identified through next-generation sequence analysis.

Statistical data analyses of the sequence reads from six controls (no interaction selection) and 36 mutant library interaction samples (each biological replicate was sequenced three times) resulted normalized interaction perturbation profiles of the NF2 isoform 7 with its four partners, respectively (Fig S2C and Table S1). The mutant NF2 libraries did not show a mutation selection bias during

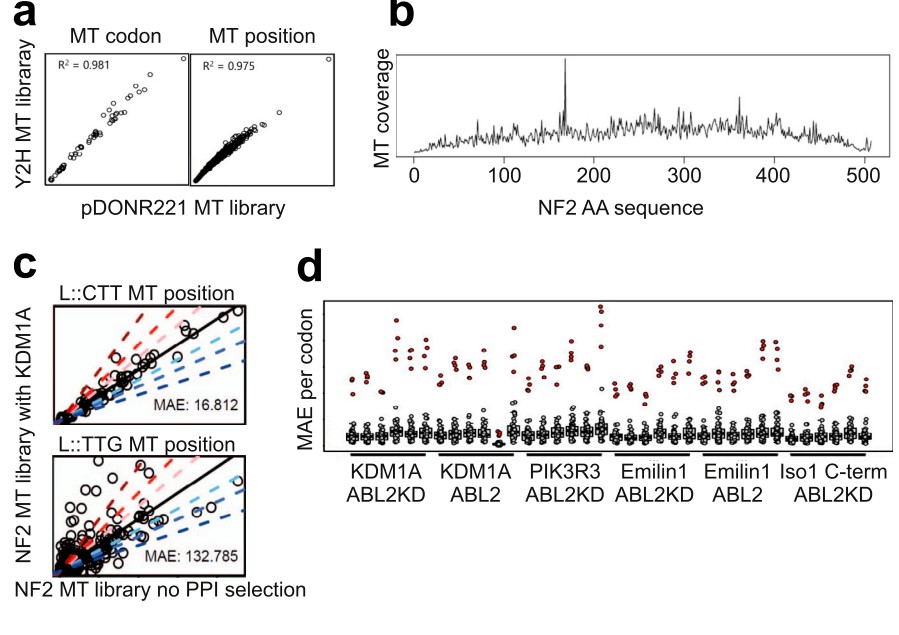

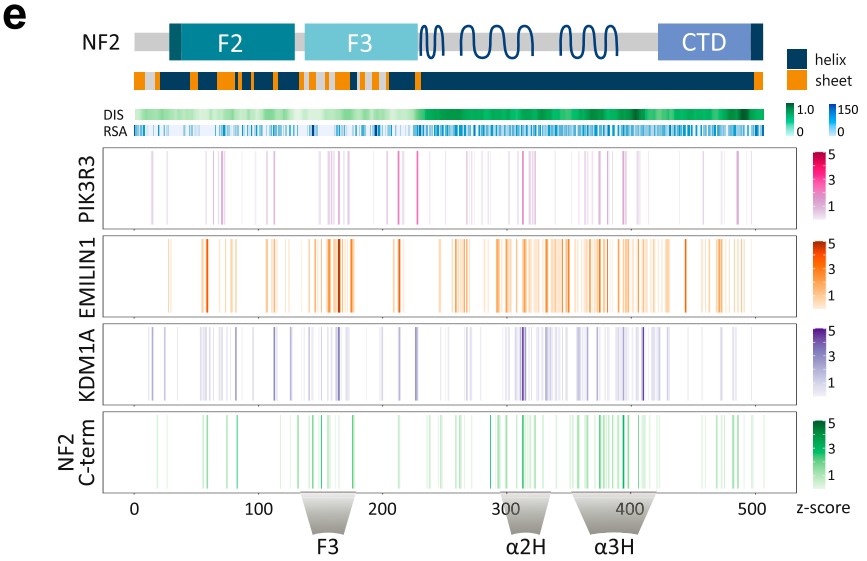

**Figure 2. Deep mutational NF2 interaction perturbation.**
**(A)** Number of mutant reads from the mutant NF2-Iso7 library in pDONR221 (library cloning vector) and pBTM116-D9 (Y2H vector). The axes represent the number each mutation was sequenced for a codon (left) or for a NF2 position (right). **(B)** Mutational sequence coverage (number of mutant reads) of the NF2 mutant library across the NF2 amino acid sequence. **(C)** Example of the recall statistics of the rY2H selection with NF2-KDM1A enrichment of a non-programmed mutant codon (L::CTT) and a programmed mutant codon (L::TTG, right) after rY2H selection of mutant library NF2 isoform 7 through interaction with WT KDM1A across all positions within NF2. Enrichment of reads in comparison with the NF2 MT library for a set of programmed (L::TTG) mutations is observed as deviation from a linear model (mean absolute error MAE = 132). **(D)** Overall results for all PPI samples. Deviation from codon-specific linear models (MAE: PPI versus NF2 library) of the programmed amino acid mutations (red dots) is much larger than of any other mutations (grey dots). MAE values of all codons for all 36 protein interactions sequenced are shown. **(E)** rY2H interaction perturbation profiles. Schematic of NF2-Iso7 protein, domain structure with structure (helix/sheet), disorder prediction (DIS), and solvent accessibility predictions (RSA). Aligned, combined enrichment profiles of the four interactions are shown. Sequence position interfering with the NF2-Iso7 interaction is color coded (Z-score). The PIK3R3 profile represents a combination of two biological replicas assayed in the presence of ABL2, the NF2 C-term profile combines two experiments with ABL2KD. KDM1A and EMILIN1 profiles contain a combination of four biological replicas (each two with ABL2 and two with ABL2KD). Mutational cluster regions are highlighted as F3, α2H, and α3H.

the cloning and strain preparation procedures (Fig 2A) and covered the whole NF2 sequence uniformly (Fig 2B). Efficient enrichment of programmed AKEL mutants through interaction selection with the rY2H system was observed, when comparing the input library with sequences obtained from the mutant library tested against a WT interaction partner (Fig 2C). For all NF2 interaction pairs, we observe a clear deviation from the codon- and position-specific linear models for the AKEL mutations but very little deviation for all other sequenced mutations (Fig 2D).

Enrichment values were normalized across the interactions and expressed as Z-score. Even though we located the respective interaction sites of the interaction partners differently to the N- or C-terminal halves of NF2, disrupting mutants were found spread over large parts of NF2 for all four interaction partners (Fig 2E). This result likely reflected both, mutations that disrupt folding and mutations linked to conformational constraints of NF2. However, three hotspot regions emerged from the interaction perturbation patterns of the interactions. Amino acid substitutions in the N-terminal half were selected perturbing the interactions in an area in the F3 part of the FERM domain. In the C-terminal half of NF2, we found a region with clustered mutations in the α2H helix (around AA 310 in isoform 7), and a second region in the α3H helix (proximal to AA 350–400 in isoform 7). The interaction perturbation profiles of the four tested interactions (Fig 2E) showed substantial overlap in three critical regions located in the N-terminal and in the C-terminal half of NF2.

## Assessing single site mutations in NF2 interactions

We next validated the deep mutational scanning results to identify mutations that selectively alter protein interactions without affecting overall protein folding and stability. 50 amino acid exchanges were individually introduced in both NF2 isoforms 1 and 7. Constructs were confirmed through sequencing and used for individually testing each interaction with the WT interaction partner proteins in a pair-wise Y2H colony matrix assay. The largest class of mutants behaved like WT NF2 (Figs 3A and S3). In addition to S518D, 15 of the single-site mutations caused a selective change of interaction patterns compared with interactions observed in their WT isoform counterparts in the pair-wise Y2H colony assay (Fig 3A). We observed distinct patterns of loss and gain of interactions with the two NF2 isoforms and its four interaction partners (Fig 3B). Five single amino acid point mutations altered the NF2 interaction with KDM1A, four with EMILIN, 15 with PIK3R3, and three substitutions modulated the NF2–NF2 interaction.

Four mutations, Q147A, S265L, R346E and W258E, showed unique patterns of interaction perturbation (Fig 3B), where W258E impacted all three PPIs. Q147A reduced PPIs with EMILIN and PIK3R3, R346E resulted in a selective loss of EMILIN interaction and the Iso1-S265L variant increased the EMILIN interaction. Interestingly, L241A, Y244L, E260L, and S265L reduced the KDM1A interaction selectively with the isoform 7, whereas the substitutions did not affect the PPIs with isoform 1. PIK3R3 did not interact with these variants, but the interaction of the four isoform 7 variants with EMILIN was not altered.

A large group of variants represented specific PIK3R3 interaction-disrupting mutations distributed across the whole protein sequence (Fig 3A, magenta). As defined in the rY2H-seq screen, the majority (A416K, E450A, L458E, Y481L) localized in the C-terminal clusters α2H and α3H. In the 384-format colony assay, the interaction of NF2 with PIK3R3 and with NF2-Iso1C appeared generally weaker. This allowed the observation that three α3H mutations, A441L, K471A, and P482E strengthened homomeric interaction of NF2 isoform 7 with the NF2 isoform 1 C-terminus (Fig 3A, green). The same NF2 variants strengthened the PIK3R3 interaction rendering it independent of ABL2-dependent PIK3R3 dimerization. Therefore, variants facilitating NF2–NF2 interaction may gradually modify the interaction with PIK3R3 as such that PIK3R3 dimerisation is less critical. Comparable with S518D, which negatively affected both the interaction with PIK3R3 and the NF2–NF2 interactions, the three α3H mutations also showed a coupled phenotype promoting both the PIK3R3 interaction and the NF2–NF2 homomerisation.

Notably, in the colony assay, all five KDM1A interaction-perturbing mutations were clustered in the F3-FERM subdomain. This is intriguing given the fact that our Y2H experiments showed that the C-terminal half of NF2 was sufficient to interact with KDM1A. The result can be explained by an indirect, conformation-driven perturbation of the interaction caused by the F3-FERM domain mutants. In summary, we identified at least two novel regions important for NF2 regulation of its protein interactions and conformation. The N-terminal F2–F3 region in the FERM domain is a key determinant to isoform interaction specificity and the α3H in the C-terminal half of the protein is involved in NF2–NF2 interactions.

## Structural impact of FERM domain mutations

Mapping the C-terminal mutations in α3H which enhanced the NF2–NF2 interaction (A441L, K471A, P482E) to a structural prediction model of full-length NF2 isoform 1 (AF-P35240-F1) revealed a localization of the residues in extended alpha helical rod-like structures of the protein, providing limited information for interpretation. However, we mapped the mutations in the N-terminal half of NF2 to a 3D atomic structure of the FERM domain solved in a closed NF2 protein state (PDB: 4zrj, [Li et al, 2015]). FERM domain structures contain three subdomains (F1, F2, and F3) forming a cloverleaf structure (Fig 4A). The F1 subdomain resembles ubiquitin, F2 has similarities to the acyl-CoA binding protein, and the F3 subdomain has structural similarities to phospho-tyrosine binding, pleckstrin homology, and Ena/Vasp Homology 1 (EVH1)-signaling domains. We note that this closed 3D structure also includes a peptide from the NF2 C-terminal part with two A585W/S518D-stabilizing mutations (chain B, 506–595) where the tryptophan 585 inserts into the F3-FERM domain (Fig 4A). In silico mutagenesis (BIOVIA Discovery Studio 2020) allowed us to model potential effects of mutations on the conformation of NF2. For example, Q147A, which had a uniquely altered interaction pattern, is located in the F2 part of the FERM domain (Fig 4A). The WT Q147 formed four hydrogen bonds (<3.4 Å), two of them connected to E152 and R198 in other helixes within the F2 domain (Fig 4B). Upon replacing Q with A in the structure, the hydrogen bonds to neighboring alpha helices were disrupted and weak hydrophobic interactions formed. Furthermore, the helix with Q147 is accessible from the site of the NF2 C-terminus.

Whereas S256 is a surface residue where the structural view is non-informative, mutations at positions Y244, L241, and W258 spatially clustered in the central part of the F3 lobe of the FERM (Fig 4A). We propose that these three variants may perturb positioning of the large alpha helix (α1F3) in the F3–F1 interface thereby affecting overall NF2 conformation. Residue Y244 was located very central in the F3 subdomain and had a hydrogen bond with amino acid L233 (~3 Å) and one hydrophobic interaction with P252 (4.2 Å) (Fig 4C). A Y244F mutation did affect these bonds, yet added weak hydrophobic interactions to Y221 (5.2 Å) and R249/L250 (4.7 Å) were observed. Modeling a Y244L mutation added weak hydrophobic interactions with R249 (5.5 Å) and L233 (5.0 Å).

The intramolecular interactions of L241 involved one hydrogen bond to F256 and four additional hydrophobic interactions with amino acids L234, V236, I273, and C300 (≥4.7 Å). The model with a L241A exchange lacks all hydrophobic interactions except L234 (5.4 Å, Fig 4D). The loss of contacts affected interactions with the neighboring β4F3-sheet and the α1F3-helix, the helix connecting the F3 subdomain with the central helical domain. We made a similar observation when modeling the W258E substitution (Fig 4D). In the 3D structure, W258 forms hydrogen bonds to residues D237, A238, L239, and I261 (~3.0 Å), hydrophobic interactions with N303/H304, L239/G240, L241, I261, and a pi–sulfur interaction with C300. A glutamic acid at position 258 lost all contacts that connected the W258-β-sheet with the α1F3-helix. The structural analysis suggested that mutations at both positions, L241 and W258, perturb the F3 domain interaction with the α1F3-helix. The α1F3-helix was shown to undergo a large structural

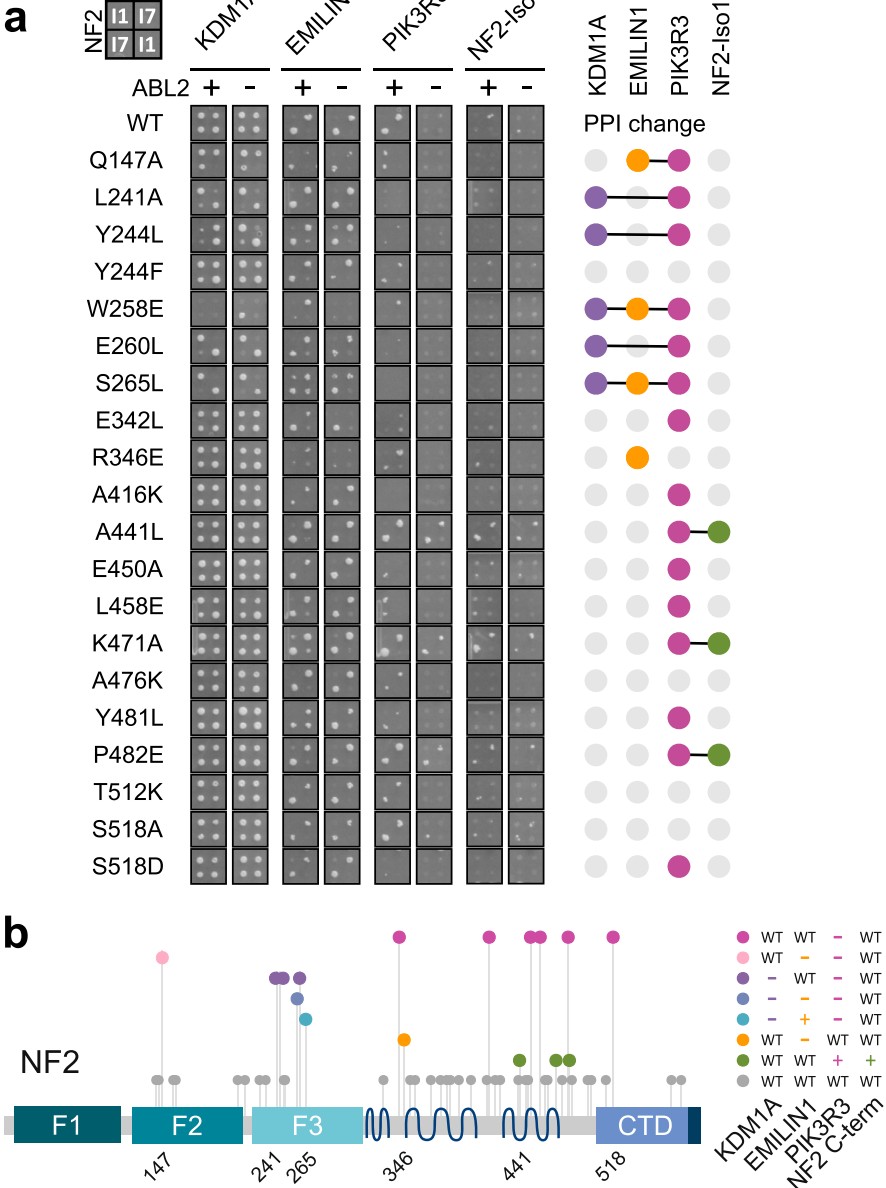

**Figure 3. Distinct patterns of loss and gain of interactions with NF2 variants.**
**(A)** Y2H protein interactions of a selected subset of NF2 single-site mutants. Selective agar, where yeast colonies indicate protein interactions, NF2 bait constructs in rows and interacting prey in columns. Each individual mutation (isoform 1 numbering) was tested as isoform 1 (384 plate format: upper left and lower right spot) and isoform 7 (384 plate format: upper right and lower left spot) in duplicate in the presence of an active tyrosine kinase ABL2 (+) or an inactive version ABL2KD (–). Right: upset plot indicating alterations of the interaction patterns in comparison with WT NF2 through colored dots. Full set of variants see Fig S3. **(B)** Mutant classes defined in the Y2H spot assay. The Y2H validation assay with single-site mutations performed in NF2 isoforms 1 and 7 resulted in specific interaction losses and gains when compared with the two WT isoforms, respectively. Projected to the protein primary structure of NF2, mutants were grouped into eight classes, including WT (grey) and four mutations with individual PPI patterns.

rearrangement upon PIP₂ and LATS binding (Chinthalapudi et al, 2018; Primi et al, 2021) (Fig 4A).The structural modeling of mutations, although not dynamic, can provide hypotheses which address these NF2 sites within the F3 domain as key residues influencing the conformational state of NF2.

E260 spatially clusters with the other interaction perturbation F3-FERM domain residues (Fig 4A); however, it does not point towards α1F3-helix, but in the opposite direction. E260 is actually at the contact interface of the F3-FERM subdomain and the C-terminal end of NF2 (Fig 4E). E260 forms a strong salt bridge to R588 localized in the C-terminus of NF2 (2.7 Å). Upon modeling a mutation to leucine at the position, this salt bridge was perturbed and replaced with a 4.9 Å hydrophobic contact. Hydrogen bonds between E260 and P257 and L276 within the F3 domain were hardly affected. For

modeling the E260L mutation, the Alphafold protein structure prediction (AF-P35240-F1) had to be used as E260 is in close proximity of the A585W stabilizing mutation in the 3D x-ray structure (4zrj) (Fig 4A). The tryptophan at 585 sterically interferes with modelling the E260L exchange; hence, this further suggests that E260 is at an important place for contacts across NF2 to the C-terminal part.

In summary, the structural analysis of NF2 variants with altered interaction patterns showed that these substitutions affected contacts to regions distant in the primary sequence, either within the FERM domain or to the C-terminal part of NF2. This observation supports the hypothesis that the sites are critical for the structural dynamics of NF2 and defines the F2-FERM domain around residue 241 and the F3-FERM domain, in particular through positions, L241,

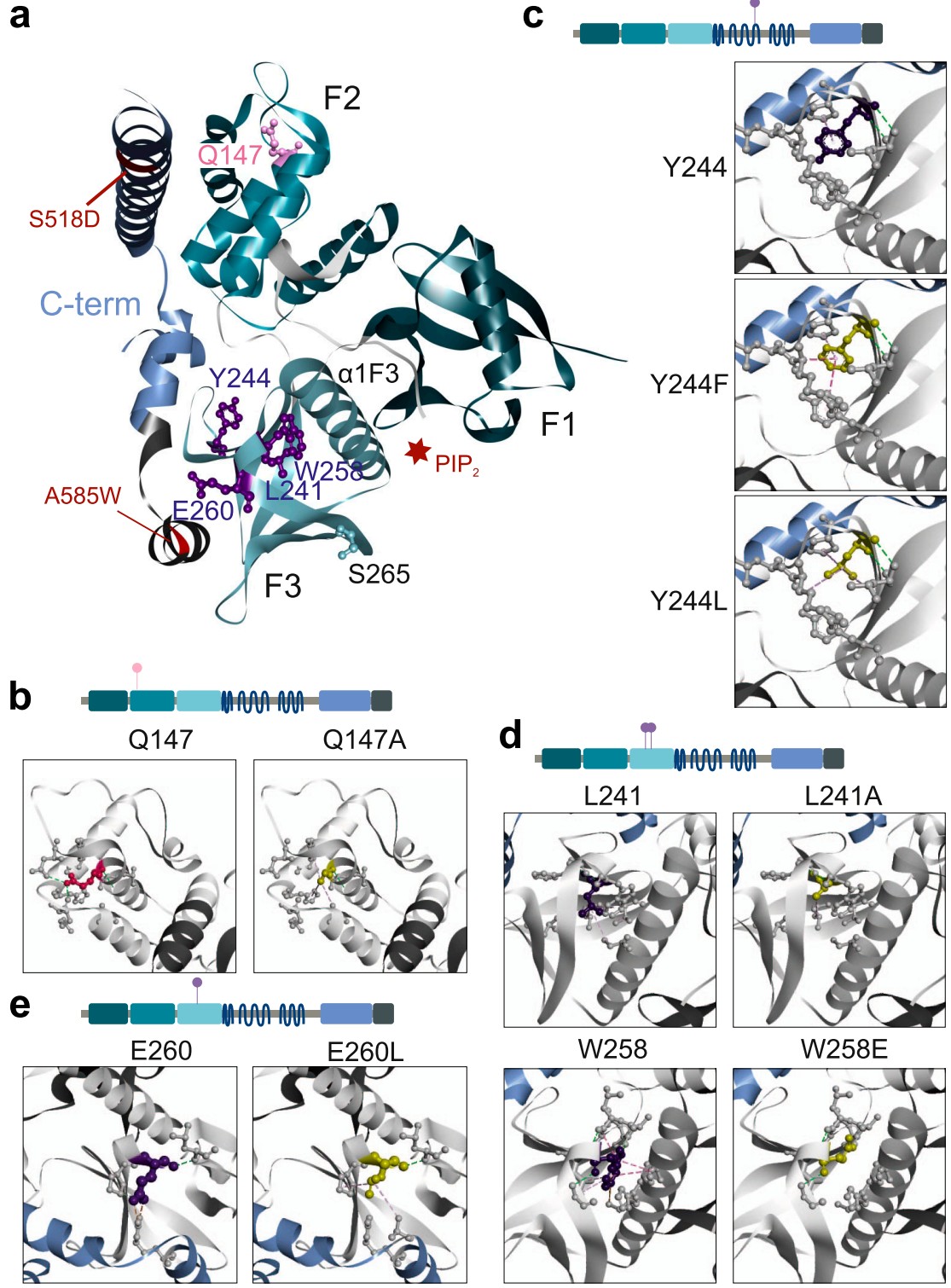

**Figure 4. Structural analyses of NF2 variants.**
**(A)** FERM domain mutations projected on the NF2 protein structure. 3D ribbon "cloverleaf" structure of the FERM domain in grey, C-terminal NF2 peptide in blue colors (PDB: 4zrj). Mutations S518D and A585W in the C-terminal NF2 peptide that stabilized the structure are indicated, and the PIP2 binding groove between F3 and F1 subdomains (red star). Amino acid positions perturbing the NF2 interaction pattern (purple, ball and stick) cluster in the F3-FERM subdomain. **(B, C, D)** 3D structures of the FERM domain as ribbon models in which indicated residues are colored and displayed as ball and stick atomic models with the respective interactions indicated as dashed lines. WT residue (left panel) in comparison with mutant residue (right panel). (B): Q147; (C): Y244, (D): L241 and W258, (E): E260. The models were created with BIOVIA Discovery Studio 2020 (version 20.1.0.19295).

W258, and E260, as the critical structure for the overall NF2 protein conformation.

### Effect of NF2 variants on cell proliferation

We examined NF2 variants for their cellular effects on proliferation in mammalian cell culture. The FERM domain is critical for sub-cellular NF2 protein localization (Brault et al, 2001). For example, isoform 7 delocalized from the plasma membrane to cytoplasmic structures (Deguen et al, 1998). A crystal structure of NF2 bound to PIP$_2$ highlighted the grove between the FERM F1 and F3 subdomains as a phospho-lipid binding site in NF2. Lipid-binding deficiency and altered protein interactions may be coupled (Chinthalapudi et al, 2018). Therefore, we first examined subcellular localization of the NF2 FERM domain mutant variants in confocal fluorescence microscopy (Figs 5A and S4). NF2 WT and the tested single amino acid variants were all found at the plasma membrane, especially at cell-to-cell contact sites. NF2-W258E is also located at the plasma membrane; however, some cells in addition showed YFP fluorescence concentrated in the perinuclear region. In conclusion, using transient transfection of YFP fused-NF2 mutant constructs in HEK293T cells, in comparison with WT NF2, we did not observe pronounced differences with respect to expression levels and subcellular localization.

We next assayed the effects of the single amino acid substitutions on cell proliferation, a key feature of the tumor suppressor function of NF2 (Sher et al, 2012; Xing et al, 2017; Chinthalapudi et al, 2018; Primi et al, 2021). FACS analysis was performed to investigate the number of transfected mutant NF2 YFP-positive cells every 24 h over a 72-h period in four experiments (triplicates each). The percentage of YFP positive cells was recorded and z-scores were calculated for each of the tested mutant constructs in each experiment and timepoint. Because apparently contradicting results on NF2 functional effects may be attributed to the use of different cell lines (Petrilli & Fernández-Valle, 2016), we tested both, HEK293 (epithelial human embryonic kidney) and A549 (epithelial lung carcinoma) cell lines. Both cell lines allowed for a relative high rate for transient transfection and are frequently used for proliferation assays.

HEK293 and A549 showed differences in their overall response to NF2 transfection. Although the fraction of HEK293 WT NF2 transfected cells (YFP positive) decreased over time, this was not observed in the A549 cell line. However, we investigated every cell line separately and assessed the effect of mutant NF2 transfection versus the WT protein. Across the 15 NF2 variants tested (S518A and S518D included), three FERM domain substitutions had the strongest and best reproducible effect on the fraction of YFP-positive cells (Fig 5B). In HEK293, Y244F showed a relatively lower YFP-positive cell number, Q147A and W258E had a relative higher YFP-positive cell fraction compared with the group of all NF2 variants. Although the A549 cell line had a higher inter-experiment variation in the FACS quantifications, Q147A, Y244F, W258E, and S265L significantly affected proliferation. Q147A, W258E showed opposite effects in the two cell lines, which may be attributed to the different genetic background and cell line-specific proliferation requirements. Consistently across the two cell lines, our analysis

showed that mutations altering NF2 conformation, specifically Q147A, Y244F, W258E, in the F2-F3 region, affected cell proliferation.

To confirm the FACS analyses, we monitored cell proliferation of YFP-tagged NF2 variants through live cell imaging using a Incucyte phase and fluorescence imaging analysis system. We transfected HEK293 cells with the WT and mutant constructs, tracked cell proliferation by quantifying cell confluency over time. In five experiments, relative cell proliferation was normalized to the average cell proliferation of all NF2-transfected cells. Confirming the results from FACS analysis with HEK293, WT NF2 reduced cell proliferation and Q147A and W285E significantly perturbed this effect (Fig 5C). Finally, we performed an analogous analysis in SC4 cells, an immortalized mouse schwannoma NF2$^{-/-}$ cell line, that has been used to study NF2 proliferation effects (Morrison et al, 2007; Cui et al, 2019). Because transfection efficiency of SC4 cells was about 10–20% only, we quantified YFP-positive cell area through live cell imaging of the NF2 variants in five experiments. Transfection of SC4 cells with NF2 Q147A and W258E significantly reduced and elevated cell proliferation in comparison with WT NF2, respectively (Fig 5D). The NF2 Q147A effect is in line with the results observed in A549 cells and W258E with HEK293, respectively. These experiments in mammalian cells demonstrate that the amino acid substitutions in the FERM domain alter cell growth in an NF2-dependent manner.

## Discussion

In analogy to ERM proteins, NF2 function and tumor suppressor activity is thought to be controlled by conformation switches between an open or a closed state. In the case of ERM proteins, the functionally activating, conformational change is mostly driven by phosphorylation of a threonine T567 residue in the C-terminal domain, a mechanism that as such does not exist for NF2 (Bretscher et al, 2002; Michie et al, 2019). Whether or not PTMs of NF2, for example at S518, cause any substantial conformational changes appears to be context specific (Surace et al, 2004; Hennigan et al, 2010; Sher et al, 2012; Ali Khajeh et al, 2014). Although NF2 conformation determines its function, it remained unclear which conformation is active and what determines the NF2 conformation upon activation (Petrilli & Fernández-Valle, 2016).

Structural data obtained with a set of protein interaction partners highlight conformationally different NF2 states induced by protein and lipid binding (Ali Khajeh et al, 2014; Chinthalapudi et al, 2018; Primi et al, 2021). For example, the NF2 interaction partner AMOT increased binding of NF2 to LATS1 (Li et al, 2015). Because the NF2 binding sites for LATS1 and AMOT were reported in the FERM domain (F2 subdomain) and the central helical domain (AA 401–550), respectively, it was concluded that AMOT binding induced a conformational opening of the FERM domain LATS1-binding site (Li et al, 2015). In vitro, LATS1 binding increased by 10fold when NF2 was preincubated with PIP$_2$, and lipid binding (in the F1-F3-cleft, Fig 4A) was suggested to release the C terminal-FERM domain intra/inter molecular NF2 contacts exposing the LATS1-binding site (Chinthalapudi et al, 2018). On the other hand, DCAF1 binding to the F3 domain had no major influence on NF2 conformation (Kang et al, 2002; Li et al, 2014; Mori et al, 2014).

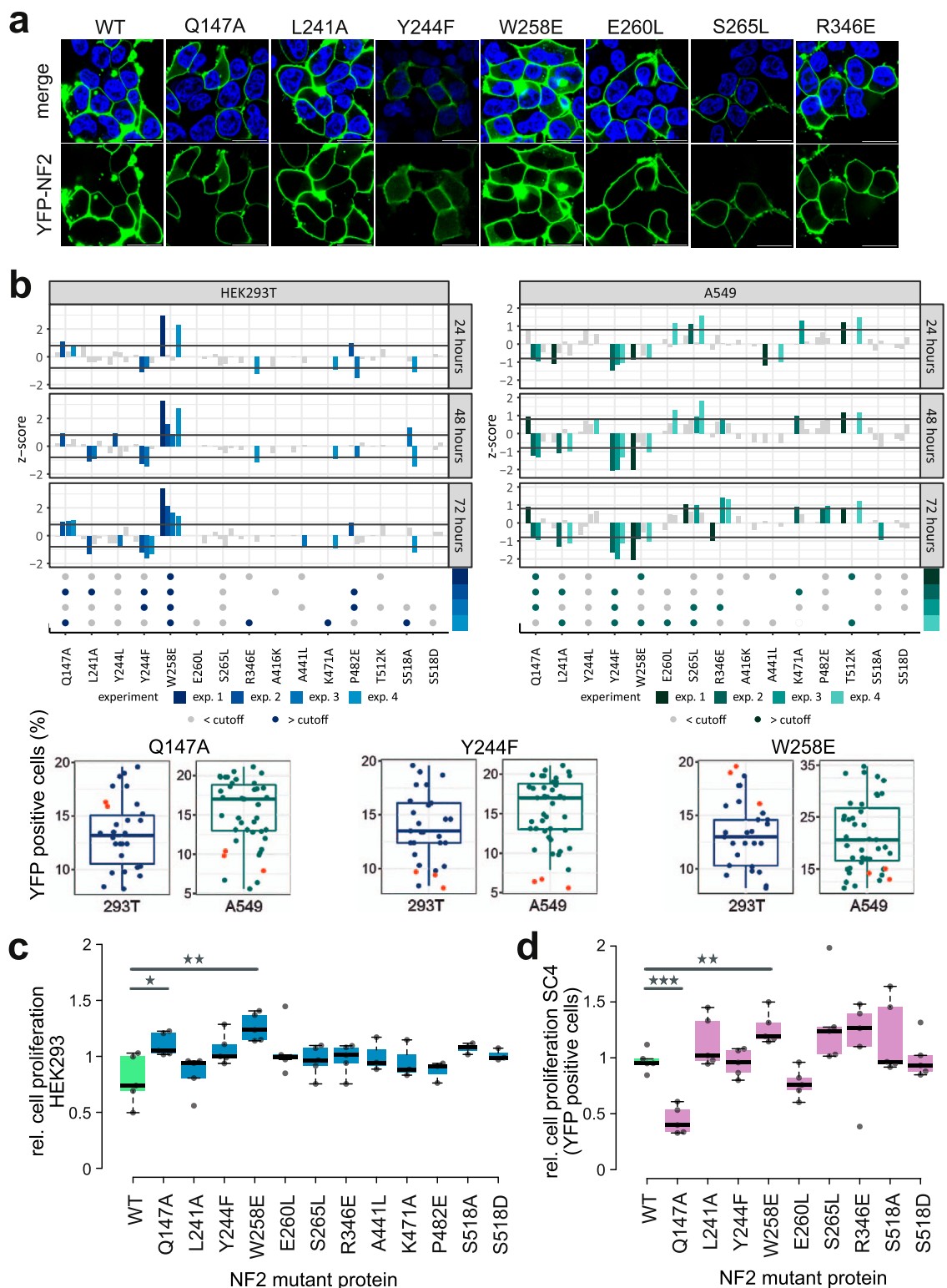

**Figure 5. Impact of NF2 variants on cell proliferation.**
**(A)** Confocal fluorescence microscopy images of WT and selected NF2 variants. HEK293T cells were transiently transfected with N-terminal YFP-tagged NF2 (green) and cellular localization of NF2 proteins 24 h after transfection was investigated by confocal fluorescence microscopy. Nuclei were stained with Hoechst (blue). Top, merge of Hoechst and YFP, bottom, YFP signal from NF2 expression at the membrane. Scale bar = 20 μM. Full set of variants see Fig S4. **(B)** FACS-based proliferation analyses of HEK293T and A549 cells expressing YFP-tagged NF2 proteins. Cells were transiently transfected with plasmids expressing YFP-tagged NF2 WT and mutant versions and the relative fraction of YFP-positive cells was determined over a time period of 3 d. Individual experiments were performed as triplicates, each NF2 mutant versions was tested in up to four experiments. Upper panel: Z-scores were calculated and variants below or above the cutoffs of −0.8 or 0.8 were colored according in each experiment.

Homomeric interactions with various NF2 constructs were observed (Grönholm et al, 1999; Meng et al, 2000). Y2H experiments and co-immunoprecipitation assays showed full-length isoform 1 dimerization with full-length isoforms 1 and 2 (Grönholm et al, 1999; Meng et al, 2000), and interaction of the C-terminus of isoform 1 and of isoform 2 (Meng et al, 2000). N-terminal NF2 (AA 1–313) was used to precipitate full-length NF2 isoform 1 in vitro, whereas the N-terminal part was not able to interact with NF2 isoform 2 (Nguyen et al, 2001). N-terminal constructs interacted with full-length NF2 and C-terminal constructs in Y2H and in vitro binding assays (Sherman et al, 1997; Grönholm et al, 1999; Sher et al, 2012). Therefore, current models of NF2 dimerization allow both N- to C-terminal antiparallel interaction and C- to C-terminal interaction. Similar to the PPI patterns with PIK3R3, our experiments demonstrated C- to C-terminal NF2–Iso7 interactions that were indirectly regulated by the FERM domain in the N-terminal part of NF2. We cannot rule out other NF2 homomeric interactions, for example, NF2 isoform 1 homodimers, which may have occurred as false negatives in our Y2H setup.

In total, 141 protein interaction partners of NF2, including KDM1A (Weimann et al, 2013; Haenig et al, 2020; Go et al, 2021) and EMILIN (Haenig et al, 2020), were previously annotated in systematic large-scale human PPI studies, most of which are not functionally characterized (Kunowska & Stelzl, 2021). Although we did not address the potential biological function of the NF2 interactions with KDM1A, EMILIN or PIK3R3 in this study, we thoroughly characterized differential isoform specific, conformation-dependent interaction patterns with these three NF2 interaction partners. The NF2 interaction partners bound to different parts of the protein, however, EMLIN, PIK3R3, and NF2-Cterm binding was negatively regulated by FERM domain structures distant in the primary sequence (Fig 1C). Our objective was to use the interactions to probe NF2 conformation in deep mutational interaction perturbation scanning experiments.

Deep mutational scanning is a powerful approach for determining the sequence–function relationships with the goal to better predict the functional consequences of genetic variation (Starita et al, 2017; Woodsmith et al, 2017), identify sites that regulate protein interaction (Starita et al, 2015; Woodsmith et al, 2017; Faure et al, 2022) or probing protein conformation and structure (Bolognesi et al, 2019). Variants of other key tumor suppressor proteins, such as PTEN and BRCA1, were successfully characterized using mutational scanning approaches coupled to functional assays (Findlay et al, 2018; Matreyek et al, 2018).

In this study, we have assessed the effect of >2,000 single amino acid substitutions in tumor suppressor NF2 on the interactions with four protein partners that bind NF2 in a conformation-dependent manner. In our interaction perturbation approach, we mutated all NF2 position to either a glutamic acid (E) or a lysine (K) introducing a negative and positive charge, respectively. We also replaced the WT amino acids with alanine (A) and leucine (L) introducing a small and a large residue, respectively. Other exchanges such as to proline, glycine, tyrosine or tryptophan were found to frequently affect stability and expression levels of the protein variants (Faure et al, 2022). On one hand, the choice of using four different amino acids instead of all possible in our experiment may limit our inferences with respect to disease variants. On the other hand, we introduce more subtle perturbations to specifically probe interactions and protein conformation (Woodsmith et al, 2017). Loss of interaction can be caused by effects on binding affinity or on protein abundance (or both) (Faure et al, 2022). We did not quantify these effects, but robustly controlled for protein folding effects in our experiments through the use of four distinct binding partners. Interaction selectivity of the perturbation effects (Fig 3) demonstrated that the studied mutations do not substantially affect protein abundance.

For all four interaction partners, we observed relatively wide spread mutation perturbation across the primary sequence of NF2 (Fig 2). The patterns are more similar than expected, and did not directly reflect larger primary binding sites of the partner proteins. Rather, both N- and C-terminal NF2 parts appeared as critical determinants for the NF2 interactions in line with our hypothesis that the PPIs reflect NF2 conformation. We observed multiple signals in the N-terminal part of NF2 when probed with EMILIN and KDM1A for which the C-terminal part was sufficient for binding. Conversely, multiple mutations in the C-terminal part affected PIK3R3 binding, for which we located the primary binding site in the N-terminal part. We selected a subset of variants with high enrichment scores (Table S1) for individual testing with the aim to identify specific interaction perturbations from different regions that do not perturb protein folding or expression. In our validation experiments, we tested 50 selected amino acids substitutions individually and a large group of variants did not show altered interaction pattern in the pair wise colony assay (Fig S3). This can be explained by the different sensitivity in the deep scanning screen compared with the pair-wise colony assay. The scanning approach quantifies enrichment of variants in a pooled, large-scale reverse Y2H experiment. The deep mutational scanning result provides a useful catalog of functional candidate variants. The pair-wise Y2H assay with individually cloned variants results a highly reproducible binary readout that does not reflect all quantitative changes from the screen, however is important for in depth functional studies.

In addition to the phospho-mimicry S518D version, 15 NF2 variants showed altered PPI pattern (Fig 3). Critical mutations clustered

---

Dots: constructs which exceeded the cutoff on two individual timepoints in one experiment were colored in blue, variants below the cutoff are colored in grey. Lower panel: examples of individual experiments where the fraction of YFP-positive cells at 48 h is shown. Data points of the Q147A, Y244F, and W258E NF2 variants are shown as red dots, respectively, blue dots show the distribution of all other NF2 protein variants in the experiment. **(C)** Live cell imaging of HEK293 transiently transfected with N-terminal YFP-tagged NF2 variants. Relative cell proliferation is calculated as area of cell confluency over select time intervals between 16 h and 64 h after transfection and normalized to the median cell proliferation of all NF2 constructs. Box plots represent the average of triplicate measurements of five experiments. Statistically significant differences ($t$ test, * = $P < 0.05$; ** = $P < 0.01$) to WT NF2 are indicated. **(D)** Live cell imaging of SC4 NF2$^{-/-}$ cells transiently transfected with N-terminal YFP-tagged NF2 variants. Relative cell proliferation is calculated as the area of cell confluency of YFP-positive cells over select time intervals between 12 h and 40 h after transfection and normalized to the median cell proliferation of all NF2 constructs. Box plots represent the average of triplicate measurements of five experiments. Statistically significant differences ($t$ test, ** = $P < 0.01$; *** = $P < 0.001$) to WT NF2 are indicated.

in two regions (overview in Fig S5). The F3-FERM subdomain harbored all five mutations modulating the KDM1A and, with the addition of R346E, the EMILIN interactions in the colony assay. In addition to the F3-FERM domain regions which appeared key to NF2 conformation, other clusters of substitutions in the α2H and α3H affected the PIK3R3 interaction. We showed in our pY-Y2H studies that PIK3R3 interacted as a dimer with NF2. Three mutations in the αH3 helix (A441L, K471A, P482E) that also relieved the requirement for PIK3R3 dimer formation promoted the NF2–NF2 C-terminal interaction. Therefore, we identified novel single-site determinants, other than S518, important for NF2–NF2 interactions.

Studying the FERM domain mutations on the 3D domain structure (Fig 4) illustrates that positions Q147, in the F2 domain, and Y244, L241, W258, and E260 form contacts bridging different secondary elements of the FERM domain most notably to helix α1F3. Modeling the respective mutations disrupted several of those contacts impacting the dynamics of the FERM domain and thus potentially allowing for large-scale conformational changes. Indeed, the Q147A, Y244F, and W258E NF2 variants most strongly affected cell proliferation in FACS and live cell image analyses (Fig 5). This result was consistent across three epithelial cell lines, HEK293, A549, and SC4; however, the effects appeared context dependent. NF2 Q147A showed increased repression of cell proliferation in comparison with WT in A549 and SC4 cells, whereas W258E showed a decreased suppression of cell proliferation in HEK293 and SC4 cells. Different effects may—for example—be because of different endogenous NF2 levels, other endogenous NF2 interaction partners, the interplay with other tumor suppressor protein activities (e.g., PTEN) or distinct dependencies on growth pathways in the three cell lines. NF2 impacts on a variety of oncogenic pathways such Rac signaling and the MAPK, AKT, YAP/Hippo, and cGAS-STING pathways and effects of NF2 variants on proliferation may depend on cell type-specific growth requirements.

In addition to functional differences associated with NF2 versions lacking exons 2–3 (Giovannini et al, 1999; Luo et al, 2015), several patient-derived NF2 related-schwannomatosis NF2 missense mutations within the N-terminal FERM domain were found which impaired NF2 interactions with DCAF1 (Li et al, 2010). Missense L46R, L64P or L141P mutations in the FERM domain were shown to convert NF2 into a loss of function phenotype, suppressing innate DNA sensing and STING-initiated antitumor immunity (Meng et al, 2021). 430 NF2 missense mutations were deposited in the archive of human genetic variants and interpretations of their significance to disease, ClinVar (19 Dec 2021) (Landrum et al, 2020). 406 were annotated as variants of uncertain significance, 11 as conflicting, and only 13 as either likely benign, likely pathogenic or pathogenic in NF2 related-schwannomatosis. Variants of uncertain significance included Q147P, Y244C, W258G, S265L, R346S/K, A416V, A441P, L458R, Y481C, and P482R/L, amino acid residue positions for which we provide multiple evidence that substitutions are likely to impact NF2 conformation and function (Fig S5). Therefore, our mutagenesis data aids NF2 variant interpretation. However, our deep mutational scanning approach tested the impact of single-amino acid substitutions on conformation-dependent interactions in yeast only. Functional interpretation of variants is therefore limited and does, for example, not account for mammalian PTMs or effects on

subcellular localization (Cole et al, 2008; Mani et al, 2011). Nevertheless, a set of NF2 variants was subjected to validation and functional testing providing novel mechanistic insight. Our data reveal two functional important regions for NF2 conformational dynamics. The α3H helix in the C-terminus mediates NF2 homomeric interactions which are critical for NF2 activity. The FERM domain, specifically the F2-F3 part, appears as the key trigger for conformational regulation of NF2 suppressor function. Our variants provide useful genetic tools for further mechanistic studies of context-dependent NF2 function.

# Materials and Methods

### Y2H colony matrix experiments

Y2H experiments were performed as described by Worseck et al using the 96 or 384 matrix format, respectively (Worseck et al, 2012).

### Mutagenic library preparation

The mutagenic library of NF2 was generated using a multi-step PCR-based deep mutagenesis approach with on-chip synthesized oligonucleotides from Custom Array, Inc. A total of 2145 NF2 primer sequences encoding NF2 single amino acid substitutions were synthesized. Each single amino acid was exchanged to alanine (A::GCT), lysine (K::AAA), glutamic acid (E::GAA), and leucine (L::TTG). The mutagenesis protocol was essentially carried out as in Kitzman et al (2015), with adaptations described in Woodsmith et al (2017).

### Interaction perturbation reverse Y2H screen

Reverse Y2H strains that can be used to select noninteracting protein variants from complex genetic libraries were used for interaction mating [RPrey_S3: MATα, his3-Δ200, trp1-901, ade2, leu2-3, 112, gal4, gal80, can1, cyh2, met2, ura3::KanMX::(lexAop)$_8$-GAL1TATA-lacZ, LYS2::(lexAop)$_4$-HIS3TATA-HIS3, met2::((LexAop8::TetR)$_2$), ho::(LexA8::TetR)$_2$-ura3; RBait_S3: MATa, his3-Δ200, trp1-901, ade2, leu2-3, 112, gal4, gal80, can1, cyh2, met2, ura3::(lexAop)8-GAL1TATA-lacZ, LYS2::(lexAop)$_4$-HIS3TATA-HIS3, met2::(TetO5A::ADE2), ho::(LexA8::TetR)$_2$-ura3]. The RBait_S3 MATa strain was transformed with the mutant libraries according to a lithium acetate standard protocol. Before mating, yeast transformed with the WT-interacting protein plasmid (RPrey_S3, MATα) was grown in nonselective medium for 12–18 h at 30°C to an OD$_{600}$ of 1–2, and then concentrated through centrifugation to reach a total of amount of 40–80 OD$_{600}$. Yeast containing mutant library DNA were collected in 1x nitrogen base (NB) media and mixed with yeast containing WT-interacting protein at 2:1. The mixture was transferred to YPDA agar plates (six-well plate format) and incubated at 30°C. After 24 h, yeast was collected in 1x NB media, diluted, and transferred to diploid nonselective NB-agar (on BioAssay 22 × 22 cm square dishes). After incubation at 30°C for 48 h, yeast was collected in 1x NB media, diluted to an OD$_{600}$ of 0.2, and equally distributed on NB-agar–containing amino acids and nucleic acids for reporter gene selection. Kinase expression was induced by addition of 200 µM copper sulfate

(7758-99-8; Merck KGaA) in the media. After incubation at 30°C for 48–96 h, colonies were collected the plasmid DNA was isolated purified through phenol extraction and ethanol precipitation. PCR was performed with a proof-reading KOD polymerase (71086; Sigma-Aldrich) using vector-specific primers.

## Sequence data analysis

Preparation (NextSeq High, 318 Cycles) for sequencing on an Illumina NextSeq 500 in a 150-base-pair paired-end read mode was done at a sequencing core facility of the MPIMG. Data analysis was performed using custom-made Perl and R scripts (Woodsmith et al, 2017). In brief, fastq files were converted to unique-paired end sequence fasta files, followed by sequence alignment using STAR alignment software in the paired-end mode (Dobin et al, 2013). Uniform distribution of mutations in the sequence and initial statistical analysis was evaluated. A linear model was used to calculate the enrichment of individual mutations and a score was calculated for each position:

$$Enrichment\ Score = \frac{observed\ total\ sequences\ for\ codon\ x}{expected\ total\ sequences\ for\ codon\ x}.$$

High-confidence cut-offs of 50 sequences and enrichment above twofold variation of the linear model were applied for coded mutations. In addition to the previously published code, refinements in the code were added. These included the following corrections: Reads with a high proportion of secondary mutations (>0.8 of the maximal signal) present in combination with a given mutant were excluded, and insertions and deletions were locally removed from the sequences and confidence cutoffs implemented (<0.7 of the maximal signal).

## Cell culture

HEK293T (ACC 872; DSZM) cells were cultured in (DMEM, 41966-029; Thermo Fisher Scientific Inc.) with 10% (FBS, 10270-106; Thermo Fisher Scientific) and 1% Pen-Strep (10,000 U/ml, 15140-122; Thermo Fisher Scientific Inc). A549 (courtesy of ZMF cell line collection, MedUniGraz) cells were cultured in DMEM/F-12 Nutrient Mixture (1:1) (31330-038; Thermo Fisher Scientific Inc.) with 10% (FBS, 10270-106; Thermo Fisher Scientific Inc.) and 1% Pen-Strep (10,000 U/ml, 15140-122; Thermo Fisher Scientific Inc). Both cell lines were maintained in a humidified incubator at 37°C with 5% $CO_2$.

## Confocal fluorescence microscopy

24 h before transfection, $5 \times 10^4$ HEK293T cells were plated in each well of a 24 well plate ($\mu$-Plate 24 Well Black ID 14 mm, 82426; ibidi). The cells were transiently transfected with N-terminal YFP-fused NF2 WT and mutant proteins using polyethylenimine (PEI, 1 mg/ml, 9002-98-6; Alfa Aesar) at 1:5 ratio of pDNA:transfection reagent. After 7 h, the medium was gently replenished. 24 h post-transfection, the cells were washed with PBS, followed by staining of the nuclei with Hoechst 33342 (20 mM, 12.3 mg/ml, 62249; Thermo Fisher Scientific) at a final concentration of 1 $\mu$g/ml according to the manufacturer's instructions. Fluorescence microscope images were acquired by using a STELLARIS 5 Cryo Confocal Light Microscope (Leica) equipped with a HC PL APO 63x/

1,40 OIL CS2 objective. For excitation, a 405 nm (Hoechst 33342) and a 514 nm (EYFP signal) laser were used. The emission range for the channels was set to 420 nm—505 nm (Hoechst 33342), and 545 nm—625 nm (EYFP signal). Image analysis was performed with the Leica Application Suite X (LAS X, version 3.5.5.) software package.

## FACS experiments

For FACS-based proliferation analyses, $9 \times 10^3$ HEK293T cells and $7–9 \times 10^3$ A549 cells were plated in each well of a 96-well plate and incubated for 24 h before transfection. The cells were transiently transfected with 150 ng plasmid DNA of YFP-fused NF2 WT and mutant constructs using PEI (1 mg/ml) at a ratio of 1:5 and 1:4, respectively. Individual experiments were carried out in triplicates. Complete growth medium was exchanged 12 and 48 h posttransfection. Cells were trypsinized (Trypsin-EDTA, 0.25% [25200-056; Thermo Fisher Scientific Inc.] and 0.5% [15400-054; Thermo Fisher Scientific Inc], respectively) according to a standard protocol and resuspended in 70–100 $\mu$l (according to cell density) ice-cold 1 mM FACS buffer for HEK293T and 2.5 mM FACS buffer for A549 cells (FACS buffer: 1–2.5 mM EDTA, pH 8.0, and 2% FBS in 1 x PBS). In four experiments, the relative fraction of transfected YFP-positive cells was recorded every 24 h over a period of 3 d. Before starting the flow cytometry measurements performed on a BD FACS Fortessa or a Cytek Aurora flow cytometer, the cells were stained with 1 $\mu$l of propidium iodide (PI [195458; MP Biomedicals, LLC], 1 mg/ml in DMSO [D8418; Sigma-Aldrich]) to distinguish dead cells. The percentage of YFP-positive cells was normalized across the 15 variants and expressed as Z-score. A cutoff of –0.8 or 0.8 was set. The following laser and filter wavelengths were used for the flow cytometry measurements; PI fluorophore: 561 nm, 670/30 nm bandpass filter; and YFP fluorophore: 488 nm, 510/20 nm bandpass filter. The flow rate was adjusted to <1,000 events/second.

## IncuCyte live cell proliferation assays

HEK293T, A549, and SC4 cells were cultured in DMEM supplemented with 10% FBS at 37°C and 5% $CO_2$ atmosphere. Lipofections were performed using jetPRIME (VWR 89129-920; Polyplus) or Lipofectamine 3000 (L3000001; Thermo Fisher Scientific) reagent according to the manufacturer's instructions. Cells were seeded in flat-bottom, transparent 96-well plates at ~3–4 × $10^3$ cells per well. 24 h post seeding, the HEK293T cells were transfected with 50 ng/well and the SC4 cells with 100 ng/well of the respective DNA constructs. To reduce cytotoxicity, the medium was changed 4 h post transfection. Starting at 24 h post transfection, live-cell proliferation and expression of YFP-tagged proteins were tracked for 48 h using an Incucyte S3 device (Sartorius).

# Data Availability

Raw data are accessible via ENA Project Accession Number PRJEB57973 at https://www.ebi.ac.uk/ena/browser/home.

# Supplementary Information

# Acknowledgements

We thank Nouhad Benlasfer and Natalia Kunowska for help with the experiments. We thank Bernd Timmermann and the members of MPI-MG Sequencing Facility for performing the second generation sequencing experiments. SC4 cells were obtained as courtesy from Helen Morrison (Leibniz Institute on Aging, Jena). NF2 isoform cDNAs were obtained as courtesy of Michael Kressel (University of Erlangen). The work was funded by the Austrian Science Fund (FWF, project P30162). The work was supported by the FWF doc.fund Molecular Metabolism (DOC 50) and the Field of Excellence BioHealth -University of Graz. E Stefan was supported by grants from the FWF P30441, P32960, P35159, and the Tyrolean Cancer Society.

## Author Contributions

CS Moesslacher: data curation, formal analysis, validation, investigation, visualization, methodology, and writing—original draft.
E Auernig: validation, investigation, visualization, and methodology.
J Woodsmith: data curation, formal analysis, software, investigation, visualization, and methodology.
A Feichtner: validation and investigation.
E Jany-Luig: validation and investigation.
S Jehle: investigation.
JM Worseck: investigation.
CL Heine: investigation.
E Stefan: resources, supervision, funding acquisition, methodology, and writing—review and editing.
U Stelzl: conceptualization, resources, supervision, funding acquisition, visualization, methodology, and writing—original draft, review, and editing.

## Conflict of Interest Statement

The authors declare no competing interests. E Stefan is co-founder of KinCon biolabs.

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
