## [Reviewer comments · Life Science Alliance]

Life Science Alliance

Mutational scanning reveals FERM domain residues critical for tumor suppressor NF2 conformation

Christina Moesslacher, Elisabeth Auernig, Jonathan Woodsmith, Andreas Feichtner, Evelyne Jany-Luig, Stefanie Jehle, Josephine Worseck, Christian Heine, Eduard Stefan, and Ulrich Stelzl
DOI: <https://doi.org/10.26508/lsa.202302043>

Corresponding author(s): Ulrich Stelzl, University of Graz

Review Timeline:

Submission Date:	2023-03-16
Editorial Decision:	2023-03-16
Revision Received:	2023-05-21
Editorial Decision:	2023-05-23
Revision Received:	2023-05-24
Accepted:	2023-05-24

Transaction Report:

Please note that the manuscript was reviewed at Review Commons and these reports were taken into account in the decision-making process at *Life Science Alliance*.

March 16, 2023

Re: Life Science Alliance manuscript #LSA-2023-02043

Prof. Ulrich Stelzl
University of Graz
Universitätsplatz 1
Graz A-8010 Graz
Austria

Dear Dr. Stelzl,

Thank you for submitting your manuscript entitled "Missense variant interaction scanning reveals a critical role of the FERM-F3 domain for tumor suppressor protein NF2 conformation and function" to Life Science Alliance. We invite you to submit a revised manuscript, according to your Revision Plan.

Thank you for this interesting contribution to Life Science Alliance. We are looking forward to receiving your revised manuscript.

Sincerely,

B. MANUSCRIPT ORGANIZATION AND FORMATTING:

Dear Life Science Alliance Editors,

This is the submission of our revised manuscript #LSA-2023-02043 by Moessler et al. entitled “Missense variant interaction scanning reveals a critical role of the FERM domain for tumor suppressor protein NF2 conformation and function”. Thank you for accepting our Reviews Commons revision plan which we carried through and finished now.

The revised version includes several pieces of new/additional data:

- *) the Y2H pair wise test, covering all 50 mutations in Isoform 1 and 7 in full as Supplemental Figure S3 (Reviewer 2)
- *) confocal microscopy images showing the membrane localization of NF2 variants as new Figure 4a and Supplemental Figure S4 (Reviewer 1)
- *) Additional experiments with the Iso1N and Iso7N fragments in Supplemental Figure S1b (Reviewer 2)
- *) revised Figure 4a including the Q147A variant (Reviewer 2)
- *) text revision with additional discussion points marked in grey (all reviewers)
- *) correction of typos and errors pointed out by the reviewers marked in grey

Please find a detailed point by point response with our answers to the reviewers in blue below.

Thank you for your support in publication of our work.

Best regards

Ulrich Stelzl

Reviewer #1 (Evidence, reproducibility and clarity (Required)):

The manuscript "Missense variant interaction scanning reveals a critical role of the FERM-F3 domain for tumor suppressor protein NF2 conformation and function" examines the effect of a reasonably exhaustive set of point mutations on the NF2 protein on protein-protein interactions and intra-protein interactions for two isoforms of NF2 (1 and 7), finding an interesting pattern of mutations in the region not associated with bindings nevertheless impact binding, and that this binding is sometimes dependent on the presence of kinase ABL2. Authors justify this by arguing conformation shifts in the protein, potentially regulated by phosphorylation, and with distinct conformations between isoforms 1 and 7 creating different interaction patterns, must explain the differences in binding properties. The paper specifically examines mutations to phosphomimetic (i.e., charged, so as to mimic phosphorylation) amino acid residues, with relevance for the probable biological regulation of this binding. Authors note that previous work has found inconsistent protein binding properties for phosphomimetic or phosphor-inhibiting substitutions on S518, in different conditions, which would be explained by other regulation of these conformational changes, a reasonable argument. Structural modeling of the mutants and their potential effects on a "closed" NF2 structure are intriguing and well-appreciated to support the paper's conclusions, and the paper is overall well-reasoned and convincing, and it should be published.

Concerns:

The kinase ABL2 is used to perturb NF2 phosphorylation, and this is not adequately justified. Kinases such as PAK2 (PMID: 11782491, PMID: 11719502) and PKA (PMID: 14981079) target NF2. In the methods referred to (Grossmann et al), nine tyrosine kinases were used for their screen, and while ABL2 was used in this paper and generated numerous interactions, it is not clear that ABL2 is the appropriate kinase to use here. The exhaustive use of many kinases would obviously be impractical and unreasonable for this study, but the choice of this kinase should be clearly explained.

Our experiments built on the method described by Grossmann et al. and for NF2 we took ABL2, FYN, TNK1, HYK from different PTK families in a pY-dependent screen. Only ABL2 catalyzed the interaction PIK3R3-NF2 interaction. We also used S/T kinases to search for phospho-dependent PPIs as described in Jehle et al. 2022, however in case of NF2 without success. Relevant for our results presented in the manuscript, we provide evidence that the kinase dependency does not necessarily relate to NF2 phosphorylation. We do not report phospho-dependent NF2 interaction and therefore a longer discussion on phospho-dependent screens may be misleading.

We show that the kinase promotes the PIK3R3 dimerization and we find mutations that relieved the PIK3R3-NF2 interaction from the kinase dependency. From the data we concluded that PIK3R3 homodimerisation requires kinase activity (shown in Figure S1b), however the interaction between PIK3R3 and NF2 is gradually modulated with respect to PIK3R3 homodimerisation. We tried to make this point more clear in text revisions.

Minor issues:

1. In the intro, authors write "While the other ERM protein family members do not have activities directly linked to cancer, NF2 tumor suppressor activity was initially characterized in flies and mice". While "directly" makes the statement technically true, it could be argued that ERM protein involvement is as legitimate as the tumor suppression activity of NF2 (PMID: 11092524, PMID: 24421310), and therefore the suggested contrast is slightly misleading. This has no relevance to the broader paper or its findings.

We amended the text accordingly.

2. Figure 2b: Authors state mutational coverage is fairly even across the protein, however there appears to be a notable spike around a.a. 180? This does not match any of the site mutations later found to be particularly relevant for interactions, which cluster around 250 and 450, and is therefore not a significant issue.

The reviewer very carefully observed spikes in the profile presenting mutational coverage of NF2. Indeed, the spikes do not impact the results. The linear models used to quantify the enrichment score accounts for the variation in read counts between codons and positions in the initial pool.

3. In the methods, cell concentration is at one point said to be 'concentrated to an OD600 of 40-80'. I have never seen cell concentration expressed this way. Authors no doubt grew cells to an OD between 1 and 2 and concentrated ~40-fold as is standard, and wish perhaps to avoid estimating concentrations as cell numbers, which would only be approximate and cell size-dependent? However, OD is only linear between 1 and 2 for cell concentrations. An OD above 4 simply cannot be observed, as all light would be blocked. Methodology here is sound, this is merely an unusual way of expressing things.

Yes, the reviewer is right and we have corrected this mistake.

4. Page 10: The authors point out that they cannot see any difference in the expression levels of the NF2 mutants. However, there is no quantification of the immunofluorescence signal supporting this information. Maybe a western blot could suffice this argument.

We substantially revised our immunofluorescence microscopy experiments (see next point). This includes revision of this part in the text. The western blot that led to the statement is shown here. Please see also text revision on Page 10.

[Figure removed by editorial staff per author's request]

5. It is very difficult to see the localization of NF2 mutants with the immunofluorescence images as they are very small. May be try with a 63X objective or focusing on just one or two cells or adding insets with higher magnification would allow the reader to view the details of Nf2 localization.

We followed the advice from the reviewer and recorded confocal images at 63x high resolution. Indeed results are much clearer, unambiguously showing membrane staining for all NF2 variants tested. We included a subset of the images replacing the Figure 5A, and show all variants tested in Supplemental Figure S4.

6. 5th line from the bottom on page 8: allowed to model -> allowed us to model

fixed

7. Line 8 from the top on page 12: inY2H -> in Y2H

fixed

8. Line 10 from the top on page 13: your hypothesis -> our hypothesis

fixed

Reviewer #1 (Significance (Required)):

Dear Editor,

The manuscript "Missense variant interaction scanning reveals a critical role of the FERM-F3 domain for tumor suppressor protein NF2 conformation and function" examines the effect of a reasonably exhaustive set of point mutations on the NF2 protein on protein-protein interactions and intra-protein interactions for two isoforms of NF2 (1 and 7), finding an interesting pattern of mutations in the region not associated with bindings nevertheless impact binding, and that this binding is sometimes dependent on the presence of kinase ABL2. Authors justify this by arguing conformation shifts in the protein, potentially regulated by phosphorylation, and with distinct conformations between isoforms 1 and 7 creating different interaction patterns, must explain the differences in binding properties. The paper specifically examines mutations to phosphomimetic (i.e., charged, so as to mimic phosphorylation) amino acid residues, with relevance for the probable biological regulation of this binding. Authors note that previous work has found inconsistent protein binding properties for phosphomimetic or phosphor-inhibiting substitutions on S518, in different conditions, which would be explained by other regulation of these conformational changes, a reasonable argument. Structural modeling of the mutants and their potential effects on a "closed" NF2 structure are intriguing and well-appreciated to support the paper's conclusions, and the paper is overall well-reasoned and convincing, and it should be published.

We thank reviewer 1 for carefully assessing our work.

Reviewer #2 (Evidence, reproducibility and clarity (Required)):

Summary

In this manuscript the authors describe the use of a reverse Y2H-based, systematic mutational analysis method to study effects on conformation-dependent interactions of the NF2 tumor suppressor protein. Using this approach, they identified regions important for NF2 protein interaction and homomer formation and correlated some of these with cellular proliferation and matched patterns of known disease mutations. Overall, this work provides useful insight into NF2 tumor suppressor function (by identifying amino acids critical for NF2 conformational regulation) while demonstrating the power of their mutational scanning approach.

Thank the reviewer for this assessment

Major Comments

1. Why does Figure 1b not include interaction results for the NF2-Iso1N fragment as bait? Did the authors test this?

Yes NF2-Iso1N was tested, NF2-Iso1N (aa1-332) is autoactive as bait and is therefore excluded from the analysis. We added this information in the figure legend to Figure 1b.

2. The authors state that the majority of retested interactions behaved like WT (i.e., could not be confirmed; page 7 last paragraph) and claim this may be due to a difference in sensitivity between the deep scanning screen and pair wise spot testing. This seems like a very vague justification for the differences between the two assays; also, it's not immediately clear to me that the high-throughput scanning assay would necessarily be more sensitive than the lower-throughput pairwise comparison assay. The authors should provide a bit more discussion on this and address the possibility of false positives in their deep mutational scanning assay.

We expanded on the difference of the pool DMS screen and the pair wise retest and added a paragraph in the discussion. While the quantitative information of the DMS screen is useful, we argue that pair wise retesting, though binary, is important for functional studies. (Discussion, page 14)

Minor Comments

1. Page 4, Line 3 from the bottom - should read 'three isoform-specific protein interaction partners' not 'partner'.

fixed

2. In Figure 1b, the interaction of NF2-Iso7-ex17 with PIK3R3 in the absence of ABL3 suggests that the observed kinase-dependence interaction of the NF2-Iso7 form may actually not be solely due to PIK3R3 homodimerization driven by phosphorylation. The authors should make note of this possibility in the text.

From the data we concluded that PIK3R3 homodimerisation requires kinase activity (shown in Figure S1b), however the interaction between PIK3R3 and NF2 is gradually modulated with respect to PIK3R3 homodimerisation. This is our explanation why mutations in NF2 can alleviate the requirement for ABL2 and the text was amended accordingly (page 8).

3. On page 5 the authors mention that Iso1N and Iso7N, when used as preys, did not interact with full-length NF2. I don't see this experiment in the figures, however.

We included the experiment in Figure S1b. However please note that, as stated in the text, the Iso1N and Iso7N, when used as prey do not result in interactions with NF2.

4. On Page 6 (first paragraph) the authors state that the PIK3R3 interaction was 'promoted through pY-dependent PIK3R3 homodimerization'. While this is a likely and reasonable conclusion, they haven't explicitly shown this, so they should be careful about making such a strong statement. I'd recommend saying 'likely promoted' or something similar instead.

While we demonstrate that PIK3R3 requires kinase activity for dimerisation we agree with the point raised and amended the text accordingly.

5. In Figure S2, the Iso7 / EMILIN1 interaction does not appear to be giving the expected result in the rY2H (i.e., there is strong growth under both Y2H and r2H conditions). The authors should comment on/acknowledge this.

The reviewer carefully observed growth on Agar lacking adenine (rY2H) for the Iso7 / EMILIN1. Weaker interactions do show some growth in the reverse readout, however growth is reduced and allows the selection of perturbing variants through enrichment and a second generation readout.

6. For the deep mutagenesis screen, why wasn't an ABL2 condition used for NF2-Iso1C (see Fig. S2b)?

This PPI is the weakest in terms of Y2H growth signal and because growth in the presence of tyrosine phosphorylation is somewhat reduced this condition was omitted.

7. For KDM1A and EMILIN1 the authors ran mutagenesis screens with both active and kinase dead ABL2, yet results were pooled. Were any differences observed in the effects of mutations on interaction between the two kinase conditions?

No differences were observed, which is why we pooled these conditions.

8. Why aren't the yeast plates shown for most of the unconfirmed interactions? These could still be included in the Supplementary Material.

We followed this suggestion and provide a new Supplemental Figure S3. We show the pair wise Y2H results for all variants, which together with the main Figure 3 sum up to 50.

9. On page 7, under Assessing Single Site Mutations, the authors refer to the Q147E mutation and reference Figure 3. However, Figure 3 shows only a Q147A mutation. Q147A is also referred to elsewhere. Which is the correct mutation?

Q147E is a mistake, which happened twice in the manuscript. We always used Q147A. Thanks for pointing out, the mistake was corrected.

10. Figure S3b shows the 20 mutations presented in Figure 3. The DMS row indicates that some of these did not produce perturbations in the DMS experiments. Perhaps I'm misunderstanding here, but weren't the 20 mutations shown (and 60 total mutations) selected based on activity in the DMS assay? Or did some of the ones selected correspond to mutations which not produce an effect? Please clarify in the text.

Except for the S518A and S518D which are there for comparison to the literature, all 50 mutations were selected based on the DMS assay. In the pair wise colony test, not all retested successfully. However, the pairwise retest yields a binary readout, growth or no growth, the DMS assay aims for enrichment of variants. See discussion on page 14.

11. Why was the S265 mutation not considered in the structural analysis (other than being shown in Figure 4a, it isn't discussed). Also, Q147 (in the F2 region) is discussed and shown in Figure 4b, but not shown in the larger overall structure in Figure 4a.

We addressed this issue and revised Figure 4a, that now shows the Q147A in the F2 part of the FERM domain. The S265 is a surface residue (shown in Figure 4a), so the structure is non informative. The text was amended to describe the figure changes.

12. The cell proliferation results are very difficult to meaningfully interpret. While it is clear that certain mutations do affect proliferation, consistency between different types of experiments and cell lines appears to be low.

We decided to present the data from three cell lines even so the interpretation is more complex than for example showing only the HEK293 experiments. We think this is important as the field has to deal with apparently contradicting results, likely in part because of the use of different cell models. We try to address this issue in the discussion and provide a little more detailed arguments in the revisions (page 14 to 15).

13. Perhaps a bit more discussion of the possible consequences of using yeast to study human NF2 interactions and how these might affect results would be useful (i.e., due to differences in membrane composition, cellular environment, post-translational modifications etc. between yeast and mammalian cells).

We agree and missed out on this point in the previous version. Please find an addition mentioning these limitations in the discussion (page 15).

14. Page 13, line 10 says 'your hypothesis'. Believe should read 'the hypothesis'.

Fixed

15. Page 13, line 15 refers to '15' NF2 variants showing altered PPI patterns; however, 16 were described in the manuscript.

Fixed, the sixteenth is the S518D which we tested but did not select in DMS

Reviewer #2 (Significance (Required)):

This work provides insight into how NF2 conformational changes relate to tumor suppressor function, which is particularly valuable since this area is still not well understood and published results have sometimes

appeared contradictory. In addition to the insights into NF2 biology provided, the manuscript also demonstrates the value of the deep scanning mutagenesis approach. Overall, the presented research is very solid and, assuming the comments presented above (most of which are minor) are addressed I have no trouble recommending it for publication.

I believe that the NF2 biology section will be of interest to a more specialized audience, while the general demonstration of the utility of the deep scanning mutagenesis will have broader appeal.

Reviewer #3 (Evidence, reproducibility and clarity (Required)):

Summary:

The authors developed a deep mutational scanning interaction perturbation technique, based on reverse yeast two-hybrid analysis, to identify important regions influencing conformation dependent protein function in NF2.

They test the tumor suppressive NF2 isoform 1 and a shorter non-tumor suppressive isoform 7 (lacking exons 2 and 3 and containing exon 16 instead of 17) and find three interacting proteins, KDM1A, EMILIN1 and PIK3R3 (KDM1A and EMILIN1 have been identified previously). They map binding regions of these proteins using fragments of NF2 isoforms 1 and 7 and by large-scale interaction perturbation mutation scanning.

Major comments:

The main scientific advancement in the study is the development of the deep mutational scanning interaction perturbation assay, but this message is somewhat lost in the main text of the results.

We streamlined the text and provide additional points of discussion on the perturbation scanning focus in the discussion.

The relevance of the binding protein that did not bind isoform 1 is unclear (PIK3R3) and the relevance of characterising the binding domains for three proteins with an unknown function is not made clear. Were these the only binding partners identified in the yeast screen? The use of isoform 7 as a construct is helpful to locate protein binding regions, but its physiological relevance is unclear. Does it have known expression or a known function in human cells?

We agree with the referees that “The use of isoform 7 as a construct is helpful to locate protein binding regions, but its physiological relevance is unclear.” This is exactly the point, to use a non-tumor suppressive isoform as a construct contrasting the binding behavior of the canonical isoform 1. We tried to summarize the knowledge about the non-canonical isoforms in the introduction (page2 bottom to page 3 top paragraph) as well as in Supplemental Figure 1. Unfortunately, literature information is sparse.

Minor comments:

Nomenclature should be updated in line with the new guidelines (i.e. NF2 vs neurofibromin)

We updated the nomenclature, thanks for point this out.

The two major isoforms are 1 and 2, differentiated by their C-terminal region (exon 17 or Exon 16). It would be helpful to describe protein binding regions using the amino acid numbering of the full-length transcripts throughout the manuscript, rather than using isoform 7 numbering in some sections.

We use NF2-Isoform 1 numbering through out the manuscript.

"Closeness", should perhaps be changed to closed-ness

Ok fixed

The significance of the A549 and HEK293 cell lines should be explained/indicated in the text.

Both cell lines allowed for a relative high rate for transient transfection and are frequently used for proliferation assays. We also used SC4, a immortalized mouse schwannoma NF2-/- cell line, without endogenous NF2. (page 11 first paragaph)

PPI should be expanded at first use.

Ok fixed

Results are included in the context of previous studies, but it needs to be made clearer in some places which results were found in previous studies and which were identified in the current study.

We went through the text and rechecked the positioning of the references. Hopefully this is now clearer.

Specific recommendations

1. 'NF2 (Neurofibromine 2, merlin)' -delewte 'neurofibromine' this has been deleted by HGNC

Ok changed

2. 'Genetic mutations or deletion of NF2 cause neurofibromatosis type 2,' -Replace neurofibromatosis type 2 with NF2 related-schwannomatosis and cite Legius et al Genet Med 2022

Ok changed and the respective reference was added (Plotkin SR et al. Genet Med 2022).

Referees cross-commenting

I cannot see any changes to this manucrypt. In particular the terms 'neurofibromine' and neurofibromatosis should be deleted

Reviewer #3 (Significance (Required)):

The authors developed a deep mutational scanning interaction perturbation technique, based on reverse yeast two-hybrid analysis, to identify important regions influencing conformation dependent protein function in NF2.

They test the tumor suppressive NF2 isoform 1 and a shorter non-tumor suppressive isoform 7 (lacking exons 2 and 3 and containing exon 16 instead of 17) and find three interacting proteins, KDM1A, EMILIN1 and PIK3R3 (KDM1A and EMILIN1 have been identified previously). They map binding regions of these proteins using fragments of NF2 isoforms 1 and 7 and by large-scale interaction perturbation mutation scanning.

The main scientific advancement in the study is the development of the deep mutational scanning interaction perturbation assay, but this message is somewhat lost in the main text of the results.

With this revision we tried to strengthen the deep mutational scanning message and fix the mistakes pointed at in the text.

Dr Smith and Professor Evans are experts on the molecular and clinical aspects of NF2

We thank Profs Smith and Evans for their time and effort providing an assessment as experts on the molecular and clinical aspects of NF2.

May 23, 2023

RE: Life Science Alliance Manuscript #LSA-2023-02043R

Prof. Ulrich Stelzl
University of Graz
Universitätsplatz 1
Graz A-8010 Graz
Austria

Dear Dr. Stelzl,

Thank you for submitting your revised manuscript entitled "Mutational scanning reveals FERM domain residues critical for tumor suppressor NF2 conformation". We would be happy to publish your paper in Life Science Alliance pending final revisions necessary to meet our formatting guidelines.

-please use the [10 author names, et al.] format in your references (i.e. limit the author names to the first 10)

A. FINAL FILES:

B. MANUSCRIPT ORGANIZATION AND FORMATTING:

Sincerely,

May 24, 2023

RE: Life Science Alliance Manuscript #LSA-2023-02043RR

Prof. Ulrich Stelzl
University of Graz
Universitätsplatz 1
Graz A-8010 Graz
Austria

Dear Dr. Stelzl,

Thank you for submitting your Research Article entitled "Mutational scanning reveals FERM domain residues critical for tumor suppressor NF2 conformation". It is a pleasure to let you know that your manuscript is now accepted for publication in Life Science Alliance. Congratulations on this interesting work.

DISTRIBUTION OF MATERIALS:

Again, congratulations on a very nice paper. I hope you found the review process to be constructive and are pleased with how the manuscript was handled editorially. We look forward to future exciting submissions from your lab.

Sincerely,
